



# Catchment-scale drought: capturing the whole drought cycle using multiple indicators

Abraham J. Gibson[1], Danielle C. Verdon-Kidd[1], Greg R. Hancock[1], Garry Willgoose[2]

[1]School of Environmental and Life Sciences, Faculty of Science, The University of Newcastle, Australia, Callaghan, New South Wales, 2308, Australia
[2]School of Engineering, The University of Newcastle, Australia, Callaghan, New South Wales, 2308, Australia

*Correspondence to*: Abraham J. Gibson (abraham.gibson@uon.edu.au)

**Abstract:** Global agricultural drought policy has shifted towards promoting drought preparedness and climate resilience in favor of disaster relief-based strategies. For this approach to be successful, drought predictability and methods for assessing the many aspects of drought need to be improved. Therefore, this study aims to bring together meteorological and hydrological measures of drought, along with vegetation and soil moisture data to assess how droughts begin, propagate and subsequently terminate for a catchment in eastern Australia. For the study area, thirteen meteorological drought periods persisting more than six months were identified over the last 100 years. During these, vegetation health, soil moisture and streamflow declined, however, all indicators recovered quickly post drought, with no evidence of extended impacts on the rainfall-runoff response, as has been observed elsewhere. Further, drought initiation and propagation were found to be tightly coupled to the combined state of large-scale ocean-atmosphere climate drivers (e.g. El Niño Southern Oscillation, Indian Ocean Dipole and Southern Annular Mode), while termination is caused by persistent synoptic systems (e.g. low-pressure troughs). The combination of climatic factors, topography, soils and vegetation are believed to be what makes the study catchments more resilient to drought than others in eastern Australia. The study diversifies traditional approaches to studying droughts by quantifying catchment response to drought using a range of measures that could also be applied in other catchments globally. This is a key step towards improved drought management.

## 1 Introduction

Drought is most simply defined as a deficiency in water to meet needs (Redmond 2002). However, the concept of drought is markedly more complex than this, as these 'needs' vary between sectors. For this reason, drought tends to be separated into four broad categories; meteorological, hydrological, agricultural, and socio-economic (van Dijk et al., 2013). Meteorological drought refers to a decline in precipitation, while hydrological droughts comprise of runoff deficits. Agricultural droughts are defined by declines in vegetation and conditions suitable for agricultural productivity (e.g. declines in soil moisture), in turn, leading to reduced economic activity and socio-economic drought (van Dijk et al., 2013). Further to this complexity, the processes of how meteorological drought leads to the other forms of drought is still poorly understood



(Dai, 2011; Kiem et al., 2016; Verdon-Kidd et al., 2017). This issue has a global dimension, with every continent experiencing water scarcity at some time over varying durations.

Large-scale, persistent (i.e. greater than 6 months) drought more commonly occurs in regions with a highly variable hydroclimate (van Dijk et al., 2013; Kiem et al., 2016; Verdon-Kidd et al., 2017). Such droughts impact crop production, water available for the environment, and town and industrial water supply. This leads to many social, economic and political

problems; with drought estimated to cost agriculture and related business worldwide, $US 6-8 billion a year (Below et al., 2007; Botterill and Cockfield, 2013). The Millennium drought in Australia lead to declines in agricultural output of up to 85% for some areas, while water scarcity lead to some population centers importing water (Verdon-Kidd et al., 2013). Recent droughts in Texas saw losses of up $US 7.6 billion in 2011 to its agricultural sector while water stores in California have been seriously depleted due to drought in 2012 (Chew and Small, 2014; Verdon-Kidd et al., 2017). Drought has also led to

widespread famine in African regions, such as the ongoing crisis in the Sahel, and the recent water shortage in SW South Africa (WFP, 2012; BBC, 2018). These instances highlight that the ability to predict drought and understand physical, environmental responses to drought is a pertinent issue in forming effective policies to manage and minimize negative impacts of this hazard. Further, determining which catchments are more or less responsive to drought is much needed, as this could be used to inform planning decisions around new frontiers or maintenance of existing agricultural lands.

Most drought studies examine declines in water availability from global, to continental, and regional scale studies (e.g. Dai et al., 2008; Deo et al., 2017; Gallant et al., 2013; Mpelasoka et al., 2008; Romm, 2011). However, to date, there has been little research conducted at the local catchment scale. In particular, there is a significant knowledge gap on how meteorological drought propagates throughout the catchment to influence other stages/categories of drought (van Dijk et al., 2013; Kiem et al., 2016). Existing studies have primarily focused on highly productive, arid to semiarid, agricultural zones,

such as; the Murray Darling Basin (MDB), SW Western Australia and Queensland in Australia, California and Texas in the United States, and areas along the Mediterranean Sea (Grove, 1986; Dai, 2011; Matusick et al., 2013; Romm, 2011; Verdon-Kidd et al., 2017). A key impetus of these studies has been recent or current water shortages, and the imposing threat of desertification. As a result, there is little understanding of persistent drought within higher rainfall areas, such as the eastern seaboard of Australia (i.e. region east of the Great Dividing Range), which is recognized as a separate climate entity with

respect to wider SE Australia (Timbal, 2010). This is crucial, given, that higher rainfall areas may have a greater role to play in adapting to the pressures of climate change, especially with respect to water security in agriculture.

Recently, Saft et al. (2015) found that persistent drought periods have more significant and long-lasting impacts on the catchment response to rainfall (i.e. conversion to runoff) in drier, flatter, and less vegetated catchments of Australia, while catchments in the higher relief, higher rainfall and well vegetated zone (such as the eastern seaboard catchments) may be more

resilient to long term drought. Better predictability, through characterizing drought likelihood and identification of causal mechanisms, allows for earlier implementation of management strategies to buffer from the impacts of drought (Romm, 2011; van Dijk et al., 2013). This is important given that global drought policy has seen a paradigm shift away from "drought-proofing" to management practices that allow for sustainable resource use under climate stress and encourage quick return to



pre-drought function once stress is lifted (Botterill, 2003; Wilhite et al., 2014). However, to develop these, management
strategies, baseline data that characterize how droughts begin, impact a catchment, end and how quickly the catchment
responds to this needs to be quantified (van Dijk et al., 2013).

This study aims to quantify catchment scale drought characteristics for a high relief, high rainfall catchment located
in eastern Australia using multiple indicators. Firstly, a long-term drought history is established, and the onset and termination
rates of drought and the large-scale ocean-atmosphere drivers that drive these events are investigated. Secondly, the
propagation of the iconic Millenium drought through the study catchments and the environmental response to its termination
is identified using pre- and post-drought states of vegetation and soil moisture using remote sensing and field-based datasets.

## 2 Study Site

The study area is comprised of two agricultural catchments located near the town of Merriwa, NSW, Australia (Fig.
1). The Krui River (585 km$^2$) and Merriwa River (808 km$^2$) catchments form part of the Goulburn River catchment (6450
km$^2$), which drain into the Hunter River catchment (Rudiger et al., 2007). Rainfall varies spatially across the two catchments
due to the large range in elevation. In the areas of lower elevation (350-400 m), average annual rainfall is approximately 550-
600 mm.yr$^{-1}$, while in areas where elevation exceeds 1000 m, average annual rainfall can exceed 1000 mm.yr$^{-1}$. This is
distributed evenly throughout the year (Rudiger et al., 2007; Kunkel et al., 2015). The large sale drivers which modulate this
rainfall include; the El Niño/Southern Oscillation (ENSO) during late spring, summer and early autumn, the Indian Ocean
80    Dipole (IOD) during winter and spring, while the Southern Annular Mode has varying impacts throughout the year (Risbey et
al., 2009).

The underlying geology of the area is predominantly Tertiary Basalt, with Jurassic sedimentary sequences of the
Sydney Basin exposed in the river valleys.  The dominant soil types are; Euchrozerms, Chromosols, Vertisols and Dermosols.
Due to the deep and fertile soil profiles, the area is of agricultural significance and sustains grazing and cropping; making up
85    around 75% of each catchment. (Hancock et al., 2015; Rudiger et al., 2007). Vegetation is described as cover grasses with
sparse eucalypt forest in the lower and central regions of the catchments, while dense, wet sclerophyll forest dominates the
northern slopes (Kunkel et al., 2019).

## 3 Methods and Data

This study aims to quantify and measure the many aspects of drought from onset and its causal mechanisms, to
90    propagation and then the causes of termination and the response of the catchment to drought. Fig. 2 outlines the methodology
used here as a framework for holistically understanding drought at the catchment-scale.



### 3.1 Rainfall Data

For this study, two rainfall records were used. Long-term monthly rainfall data was obtained from the Bureau of Meteorology (BoM) for two stations; Roscommon (Station No. 061287), and Terragong (Station No. 061073). The Roscommon rainfall record covers the period from 1970 to 2015 (and is 98.7% complete), while the Terragong station provides data from 1908 to 1970 (and is 98.7 % complete). The stations were found to be analogous using regression analyses (Terragong Rainfall = 1.03 x Roscommon Rainfall $R^2 = 0.93$, $p < 0.001$) and so were combined to form a long-term rainfall record (1908–2015) for the study area. This provided the long-term, historical data required for the drought index applications to create a single drought history for both catchments (referred to as the study area).

Catchment specific, instrumental rainfall data was sourced from the Scaling and Assimilation of Soil Moisture and Streamflow (SASMAS) project (Rudiger et al., 2007). A total of 13 monitoring stations are situated throughout the Krui and Merriwa catchments, with rainfall recorded using a 2 mm tipping bucket rain gauge. Rainfall has been recorded at the stations since 2005 (Rudiger et al., 2007). The Krui Catchment rainfall data was 83.2 % complete, while the Merriwa Catchment Data was 85.3 % complete. This data provided the basis to analyze rainfall distribution and trends during the Millennium Drought in conjunction with the BoM data.

### 3.2 Streamflow Data

Modelled AWAP streamflow data (run 26j) was used in this study (1901–2014) as there is no long-term streamflow data available for either catchment (Raupach et al., 2009; Raupach et al., 2012). AWAP streamflow is generated using the WaterDyn model, with water balance calculations being carried out using meteorological data (e.g. rainfall, potential evapotranspiration) and continental parameters (e.g. soil characteristics, vegetation). The raw output data is monthly average, runoff (mm) on an approximately 0.05° x 0.05°, Australia wide grid (Raupach et al., 2009; Raupach et al., 2012). Australia-wide validation of the AWAP dataset can be found in Raupach et al. (2009). A single record to match the rainfall record, using average streamflow from both catchments, was then produced for 1908–2015.

### 3.3 The Standard Precipitation Index and Surface Water Supply Index

The two most common meteorological drought indices are the Standardized Precipitation Index (SPI) and the Palmer Drought Severity Index (PDSI). Given the absence of a local instrumental, long-term temperature record for the region (a requirement for the PDSI calculation), SPI was selected to identify meteorological drought (see McKee et al. (1993) for detailed description on the SPI calculation). The SPI is an indicator of how many standard deviations that the precipitation over a defined number of months lies from the long-term average (National Drought Mitigation Centre 2017). For this study, an SPI period of 6 months (SPI6) was chosen to capture persistent droughts, with meteorological drought onset being defined as a period of six successive months of SPI6 below -1. Drought severity was then categorized, based on SPI values, as; mild ($-1 < SPI < -0.5$), moderate ($-1.5 < SPI < -1$), severe ($-2 < SPI < -1.5$) and extreme ($SPI < -2$), as recommended by McKee et





al., (1993, 1995). Following Verdon-Kidd et al (2017), termination of a drought event was then defined as a period of six, successive months where SPI6 was above -1 (McKee et al., 1993; McKee et al., 1995). Rainfall data from the composite record

of the Roscommon and Terragong weather stations was used for calculation of the SPI.

The Surface Water Supply Index was applied in this study to identify hydrological drought (see Doesken et al. (1991) for a detailed description of the SWSI calculation). Similar to the SPI, the SWSI represents how many standard deviations streamflow for a specified number of months deviates from the long-term average (National Drought Mitigation Centre 2017). A SWSI period of six months (SWSI6) was also used to capture persistent drought, with periods of hydrological drought and

drought severity being defined in the same way as for meteorological drought using the SPI6. The streamflow time-series created from the AWAP dataset was used to calculate the SWSI6 which corresponded with the SPI6 calculated for the general study area. The potential issue of non-stationarity in the SPI6 and SWI6 timeseries was tested using an adjusted Dicky–Fueller test and found to be not significant for the study area from 1908–2015 (Rashid and Beecham, 2019).

### 3.4 Rainfall-Runoff Relationship

Changes in the rainfall-runoff relationship during drought were examined by methods similar to those presented in Saft et al. (2015). Annual rainfall anomalies were calculated for the same rainfall record used to calculate the SPI6, as well as the percent change from the mean for running three-year periods. Periods of three or more years of where the anomaly were less than 15% of the mean were classed as "drought periods". This classification excludes short, less intense meteorological droughts which are unlikely to lead to hydrological drought and a change in the rainfall-runoff relationship (Kiem et al., 2016;

Saft et al., 2015). Annual runoff, calculated from the AWAP streamflow used to calculate the SWSI6 timeseries, was then normalized using a Box-Cox transformation (Box and Cox 1964) and regressed against annual rainfall. This was carried out for each "drought period" and non-drought years, with a t-test used to determine if a significant change within this relationship had occurred within each drought period (Saft et al., 2015).

### 3.5 Climate Data

The relationships between various climate drivers and drought onset are explored in this study. The climate index data used for this is summarized in Table 1. Similarly, the relationship between synoptic scale events and drought termination are also explored. This is achieved using monthly geopotential height anomalies from the 20th Century Reanalysis V2c data (Compo et al., 2011) provided by the NOAA/OAR/ESRL PSD, Boulder, Colorado, USA, from http://www.esrl.noaa.gov/psd/.

### 3.6 Normalized Difference Vegetation Index

An important indicator of water availability is vegetation health; with the normalized difference vegetation index (NDVI) being a simple remote sensing method to measure this. The NDVI is generally regarded as a measure of greenness, calculated from the difference in reflectance of the red and near infra-red bands (e.g. Peters et al., 2002, Sawada and Koike 2016, Verdon-Kidd et al., 2017). This study utilized 0.05° resolution Moderate Resolution Imaging Spectrometer (MODIS),





NDVI data. This dataset provided monthly NDVI values from 2000 to 2015 and, Advanced High-Resolution Radiometer

(AVHRR) NDVI data used to extend the temporal coverage to 1982 (http://reverb.echo.nasa.gov/reverb/). The data was then standardized to remove seasonal cycles, with values ranging from +3 to -3. Many studies exist demonstrating the ability for NDVI to detect agricultural drought and response of vegetation to drought (Deleglise at al., 2015; Vicente-Serrano et al., 2013). Recent drought conditions in Australia have highlighted the costly impacts of drought when naturally occurring pastoral feed is limited and needs to be supplemented with grain and hay (Kiem et al., 2016). Given the dependence of Australian agriculture

on native and unimproved pasture; negative values are considered to show high levels of plant water stress and reduced feed, while positive values show good plant health and feed availability (e.g. van Dijk et al., 2013; Verdon-Kidd et al., 2017).

### 3.7 Soil Moisture

The most accurate indicator of agricultural drought is field measured soil moisture, as this directly relates to the water available for plant uptake. However, a significant issue (in Australia and globally) is the lack of continuous in situ soil moisture

data (Rudiger et al., 2007). The study catchments have one of the longest, continuous running, soil moisture measurements in Australia as part of the SASMAS (Rudiger et al., 2007) project. Soil moisture (% v/v) across the two catchments has been recorded at each of the 13 stations soil at depths of 0-300 mm, 300-600 mm, and 600-900 mm. This is carried out using a vertically inserted, Campbell Scientific CS616 water content reflectometer, with a recording taken every 5 minutes, and the twenty-minute average taken from this. For this study, daily average soil moisture was used. This data extends from 2003 to

2015, and so represents conditions during and after the Millennium Drought. These datasets were, on average, 84.4 % complete.

## 4 Results

### 4.1 Drought Identification

#### 4.1.1 Defining the Drought Record

Fig. 3 presents the SPI6 values derived for the study area from 1908-2014, with periods of meteorological drought highlighted. As shown here, 13 drought events of varying severity occurred over study period (Table 2). On average, these events lasted approximately ten months, with the longest being 17 months duration and six-months the shortest. In terms of intensity, the meteorological droughts tend to be rated as 'moderate' (McKee et al., 1993; McKee et al. 1995), with few severe to extreme periods experienced (average SPI6 value = -1.36). Drought events are punctuated by periods of positive SPI with a

large change in SPI in the month of drought termination (0.51–2.04). This indicates a large amount of rainfall occurring within a month (e.g. from a monsoonal trough) leading to drought termination, which rapidly relieves drought stress on water availability and vegetation production.





The SWSI6 values for the study area from 1908-2014 are also presented in Fig. 3 with eight hydrological drought events highlighted of varying intensity (Table 3). There is strong correlation between the SWSI6 and SPI6 within the study area (r = 0.75, p < 0.001), indicating that meteorological drought quickly leads to hydrological drought. Event intensities are quite variable (average SWSI6 = -1.50), while durations range between seven to 17 months (average of 11 months). Like the SPI6, the SWSI6 is strongly positive in between drought periods and there is a greater range in the magnitude of changes in SWSI6 when drought events terminate (0.14–3.29). Again, this indicates a quick return to water surplus.

### 4.1.2 Rates of Drought Onset and Termination

Fig. 4 presents rates of drought onset and termination for the study area for each drought identified in Sect. 4.1.1 as well as the average of all events. These have been calculated as the relative difference in SPI6 and SWSI6 in the six months (to match the index lengths) prior to and after the month of drought onset or termination. Both meteorological and hydrological droughts usually exhibit a steady rate of drought onset in the six months preceding drought onset, with average rates of -0.30 ± 0.22 and 0.21 ± 0.12 respectively. Rates of drought termination are similar to onset rates, with average rates of 0.23 ± 0.16 and 0.21 ± 0.16 for the respective drought indices; however, there are sharp breaks in average slope one month prior to drought termination. The rates of onset and termination uphold the notion that drought is a "creeping" phenomenon and are terminated rapidly; making long-term prediction of drought onset, run-length and termination difficult.

### 4.1.3 Causes of Drought Onset and Termination

Fig. 5 highlights the trends for each climate mode (as represented by the indices outlined in Sect. 3.5) during the six months leading into and following drought onset for the 13 droughts identified. The index thresholds for the various climate modes to be classified as a dry, wet or neutral phase (±0.5 standard deviation) are also indicated. It shows that, in the six-months prior to drought onset, ENSO tends to be in the neutral phase, with a large degree of variability. However, in most cases there is a shift to El Niño conditions following month 0 (month of drought initiation). The number of months with index values above 0.5 is greater following drought onset, with no droughts having values below -0.5 (i.e. La Niña conditions) once onset has occurred. This pattern indicates that ENSO may not be initiating drought, but rather sustaining it. Indian Ocean SSTs (represented by the II (Table 1)) show a clear trend towards cooler SSTs in the east Indian Ocean five to one month before drought onset. In contrast to ENSO, the Indian Ocean SSTs trend towards neutral conditions post onset. Over shorter lead times (two months) SAM also exhibits a transition towards more negative values leading to drought onset and for four months on average post drought onset.

The varying timing of the three climate modes impact on drought is most likely due to the seasonal nature of each climate mode. Nine out of 13 droughts in the study area initiate (month 0) in the six-month period between September and February. Indian Ocean variability is most active during Austral winter/spring, ENSO also primarily impacts NSW during the austral winter/spring, while SAM is known to influence rainfall during spring and summer. From this, it is proposed that a



combination of cool SSTS in the eastern Indian Ocean and negative SAM sets up the drought for our catchment and El Niño
sustains it.

To further highlight this, droughts Four and Nine are presented as case studies (Fig. 6). Drought Four represents a
typical drought onset, as depicted in Fig. 4, where there is a gradual decline in SPI6 in the six months leading to drought.
During this lead-up (June-December) to drought, negative II values represent cold SSTs to the NW of Australia (and eastern
Indian Ocean), indicating a positive IOD. An El Nino event also developed at the same time with the Niño 3.4 trending positive
(0.5 < 1). It is this sustained effect of two climate drivers being in their "dry-phase" that lead to the persistence of this drought.
Drought Nine, however, is an anomalous drought in terms of onset and can be easily seen as an outlier in the SPI6 drought
onset rates in Fig. 4. This drought is well recognized for having all three drivers locked into their "dry-phase" (Verdon-Kidd
and Kiem 2009). This is shown in Fig. 5; a positive SAM during autumn-winter, leads into negative II values during winter-
spring, which is followed by an El Niño event.

Presented in Fig. 7 are the changes in SPI6, monthly rainfall totals, and, average monthly rainfall totals in the six
months before and after drought termination for droughts Four and Nine. Rainfall deficits are maintained by below average
rainfall in the six months leading to drought termination, with above average rainfall falling during the month of and month
after drought termination. This is shown in the abrupt termination of meteorological and hydrological droughts in Fig. 3 and
by the large breaks in the SPI6 and SWSI6 in Tables 2 and 3. This is consistent with the notion that droughts are most often
broken abruptly by synoptic events (e.g. Verdon-Kidd et al., 2017). Fig. 8 shows anomalies in geopotential height for the two-
months prior to, and, post drought termination for Droughts Four and Nine. Both time-series show high-pressure systems
dominating eastern Australia prior to drought termination followed by the development of a low-pressure system across the
west and south-west. During drought termination, monsoonal troughs (denoted by a 'dip' in the isobars) cause widespread,
sustained rainfall across much of the eastern Australia. These are then followed by low-pressure systems developing in northern
Australia; bringing further rainfall as reflected in Fig. 7. These synoptic patterns reflect the sharp changes in SPI6 during
drought termination and show that droughts are ended abruptly by synoptic scale events in contrast to their prolonged onset.

## 4.2 Changes in Long-Term Rainfall-Runoff Relationship

Fig. 9 presents the annual rainfall-runoff relationships during the drought periods identified using the methods in Sect.
3.4. These closely align with the longer-term drought events identified using the SPI6 and SWSI6 (Tables 2 and 3). Drought
causes a downward shift in the relationship in all drought periods; however, this shift is not statistically different to annual
rainfall runoff relationships. Following the conclusions of Saft et al. (2015) this indicates that the catchment is able to recover
from drought with no long-lasting hydrological impacts.

## 4.3 Drought Propagation and Catchment Response

Few studies explore how the effects of rainfall deficits propagate throughout a catchment (van Dijk et al., 2013; Kiem
et al., 2016). This is a significant challenge in improving understandings of drought and developing better forecasting and





drought response policy (Kiem et al., 2016). Using the Millennium Drought as a case study, catchment-wide vegetation and soil moisture responses to rainfall deficiencies were assessed to quantify drought resilience.

### 4.3.1 Rainfall Distribution

Given the persistent nature of the Millennium Drought and the widespread impacts on agriculture in eastern Australia
(Nicholls, 2004; Verdon-Kidd and Kiem, 2009; Vance et al., 2014), in this section we investigate how this major drought effected our study catchments. Fig. 10 presents annual catchment average rainfall totals for the Merriwa and Krui River catchments derived from the SASMAS data and the annual rainfall totals from the BoM Roscommon station. Average annual rainfall totals for the drought period were; 668 mm, 619 mm and 550 mm for the Merriwa, Krui and Roscommon datasets respectively, while during the non-drought period they were; 651 mm, 647 mm and 664 mm. In comparison, the long-term
average annual rainfall total for the BoM Roscommon site is 603 mm. These represent annual declines of 2–5 %. The coefficients of variance for annual rainfall, for all years of data from 2003–2015, were 36 %, 26 % and 29 % for each respective dataset, while the long-term value for Roscommon (1970–2015) was 23 %.

It is widely accepted that the Millennium Drought was strongly seasonal in nature (autumn/winter) (Verdon-Kidd and Kiem, 2009). Therefore, the annual totals may not reveal the nature of the drought impact on the catchment. The percent
change in seasonal medians during the defined drought period of 2003-2009 to the post drought period in the available catchment specific rainfall data for the Krui and Merriwa River catchments and the BoM Roscommon station are shown in Fig. 11. Also shown here is the percent change in seasonal medians during the drought period to the long-term rainfall record at Roscommon (1970–2015). All three datasets show a decrease in rainfall of 20-50 % in autumn rainfall between the drought period. Based on the long-term Roscommon record the autumn rainfall during the Millennium Drought was 56 % lower than
when compared to the long-term record. Interestingly, spring rainfall was reduced (31 %) across the Krui catchment during spring, however spring rainfall was higher by 30 % in the Merriwa catchment during the drought. These differences over a small geographic range highlights the need to assess drought characteristics at the catchment level. Summer rainfall during the drought period shows little variation from the non-drought periods for either catchment.

### 4.3.2 NDVI

Fig. 12 presents the standardized NDVI time-series for the Merriwa and Krui catchments from 1982 to 2015. Also highlighted here are the periods of meteorological and hydrological drought as defined using the SPI6 and SWSI6 (Sect. 4.1.1). NDVI is well correlated with the SPI6 ($r = 0.56$ $p < 0.001$), with this correlation not being improved by applying month-step lags. This shows the propagation from meteorological drought to agricultural drought is rapid. However, while the catchment is impacted by drought, vegetation also recovers quickly from rainfall after drought.



### 4.3.3 Soil Moisture

Fig. 13 presents daily catchment-average soil moisture time-series data from 2003-2015. Coefficients of variance for median daily average soil moisture in the Krui catchment are 28 %, 23 % and 17 % for 0–300 mm, 300–600 mm, and 600–900 mm depth respectively. Within the Merriwa catchment these are 45 %, 31 % and 20 %. Average soil moisture across the soil profile is well correlated with SPI6 ($r = 0.58$, $p < 0.001$), again, this was not improved by applying month-step lags. This indicates the impacts of meteorological drought propagate quickly through the catchment. For example, all depths, across both catchments show decreases in median soil moisture from year to year during drought events during 2004-2005 and 2009. Similarly, increases in median daily soil moisture also occur at comparable times to drought termination; the years 2007 and 2010 show strong increases in median daily soil moisture; corresponding with the end of drought events. The strong correlation with the SPI6 and declines in soil moisture during drought periods indicate that the study area is impacted by drought. However, similarly to NDVI, this correlation and small variance, indicate that the catchment responds quickly to drought termination; showing that the catchment is resilient to drought.

## 5 Discussion

### 5.1 Drought Characteristic and Causal Mechanisms

Notable protracted drought events have occurred across the wider SE Australia region over the last 80 years, including; the World War II (WWII) Drought (1937–45), the 1982–83 Drought, and the Millennium Drought (Nicholls, 2004; Verdon-Kidd and Kiem, 2009; Vance et al., 2014). The meteorological drought record established for the study area corresponds closely with records established by Verdon-Kidd et al. (2017) for the Upper and Lower MDB (Fig. 1) with respect to timing and run lengths of the droughts. Drought is however less severe in the study area, with an average SPI of -1.36, compared to -1.74 and -1.61 in the Upper and Lower MDB respectively (Verdon-Kidd et al., 2017). It is proposed that the similarity in the drought records run lengths is linked to the large-scale ocean-atmosphere drivers driving climate over eastern Australia and by establishing these records and comparing onset and termination rates with climate data, drought predictability can be improved.

Not all meteorological droughts were found to progress to hydrological drought for our study catchment. The best example of this is the absence of a hydrological drought with the 1982–83 meteorological event. This was the most severe meteorological drought, with an average SPI6 of -2.32, and had a rapid rate of onset (outlier in Fig. 3), however, there was no associated hydrological drought. Rather, protracted droughts, such as the Millennium and WWII events, tend to lead to hydrological drought. This can be attributed to the time required for the effects meteorological to drought to propagate through a catchment and cause hydrological drought (Leblanc et al., 2009; Leblanc et al., 2012). Evidence for this is shown in Fig. 3, with average onset rates for hydrological drought being lower than those for meteorological drought. Similarly, the spread of drought onset rates is also lower for hydrological events than meteorological events. Overall, this indicates that long-term





periods of repeated droughts pose a greater threat of generating hydrological drought, as reduction in water availability propagates gradually throughout a catchment's water cycle. This is consistent with the notion of drought as a "creeping" natural disaster.

Many studies have investigated the causes and nature of protracted droughts in the Australian and global contexts (Gallant et al., 2013; Nicholls, 2004; Verdon-Kidd and Kiem, 2009; Vance et al., 2014). What is shown here is that the most severe droughts in this high relief catchment are not attributable to the influence of a single climate driver, rather they occur as a result of multiple modes locked into their respective dry phases across different seasons. Long-term rainfall forecasts are limited globally, proving a significant barrier to predicting long-term droughts and their cessation). However, a clear link is established here, and supported within the literature, between multiple climate drivers and drought at a large scale (Gallant et

al., 2013; Mpelasoka et al., 2018; Verdon-Kidd and Kiem, 2009). Also, this result highlights that drought predictability could be improved by improved forecasting of the dominant climate drivers that lead to drought (Mera et al., 2018).

     In contrast to drought onset, both meteorological and hydrological drought seem to terminate rapidly in the study area. Dettinger (2013) and Verdon-Kidd et al., (2017) suggest that drought termination is more closely linked to synoptic scale processes rather than the large-scale climate drivers associated with drought onset (e.g. ENSO, IOD). This is consistent in Fig.

3 for both meteorological and hydrological drought, with nearly all events showing sharp breaks in slope one month prior to drought termination. Events such as East Coast Lows, polar storm fronts and monsoon troughs are closely linked with drought termination along the eastern seaboard and we demonstrate clearly the role of monsoon troughs in breaking droughts in our study catchments. These manifest as higher than average rainfall (Fig. 6) and sharp breaks in SPI6 and SWSI6 upon drought termination (Verdon-Kidd et al. 2017). Similarly, to drought onset, understanding and improving the predictability of these

systems can improve drought forecasting (Mera et al., 2018).

## 5.2 Drought Propagation and Catchment Response

     Both study catchments displayed a strong decrease in autumn rainfall during the Millennium drought. This deficiency in autumn rainfall is a well-observed characteristic of the Millennium Drought; and the climate drivers causing drought for the study area indicate it has been impacted by similar drought mechanics seen across wider Australia (Gallant et al., 2013; Vance et al., 2014; Verdon-Kidd and Kiem, 2009).

et al., 2014; Verdon-Kidd and Kiem, 2009). However, when compared to inland regions (west of the Great Dividing Range) of SE Australia, declines in annual and seasonal rainfall are far less severe (Verdon-Kidd and Kiem, 2009). This reinforces the results of the SPI6 analysis showing that the catchment is affected by droughts similarly to surrounding areas, however, less severely.

     The strong concurrent correlation of NDVI with the SPI6 is indicative of how quickly the catchment responds to

variations in rainfall. Decreases in NDVI, indicate water-stress associated with the onset of drought and a subsequent reduction in available feed (Deleglise et al., 2015, Sawada and Koike 2016, Verdon-Kidd et al., 2017). During the severe-extreme drought events of 2002–03, 2005–06, and 2009, significant water stress was shown through the strong negative NDVI anomaly values. However, NDVI anomaly values returned to positive values quickly when drought conditions eased, indicating pastoral feed



quickly becoming available. In addition to this, variance in NDVI values across the entire time-series were small. Considering

that drought magnitude, based on the SPI6, was lower than in other areas (Verdon-Kidd et al., 2017) during the Millennium Drought, the recovery rates and small variance indicate resilience within the two catchments. We argue that understanding vegetation response to drought is important for effective land management. Characterizing drought propagation in this way allows decision makers to develop strategies to plan, long-term for drought and reduce the need for financial assistance and environmental damage.

Declines in soil moisture in response to meteorological drought are also associated with the onset of hydrological and agricultural drought (Leblanc et al., 2009; Lodge and Johnson 2008). Soil moisture is similarly correlated with the SPI6 with no lag; indicating again a quick catchment response to meteorological drought onset. While the link between precipitation and soil moisture can be hard to quantify due to the episodic nature of rainfall variability and the continuous nature of field measured soil moisture variability (Lodge and Johnson 2008), the analysis presented here indicates that the drought has

affected soil moisture availability across the study area. However, the variation is small over the time-series and decreases with depth. Soil moisture is important for regulating vegetation growth and rainfall runoff relationships. The reliability of orographic rainfall here acts as a buffer, reducing the impacts of drought and allowing for pre-drought conditions to be restored quickly after rainfall, with deep water still available during the drought due to reliable orographic rainfall. This is consistent with the notion that high-relief, well vegetated catchments may be buffered or less impacted by drought (Petheram et al., 2011;

Potter et al., 2011; Saft et al., 2015).

This idea is explored further in the stability in the rainfall-runoff relationship. Saft et al. (2015) found that persistent drought can lead to a long-term shift in the rainfall-runoff relationship. Across 139 % catchments in eastern Australia, only approximately 20 % showed no decrease in the rainfall-runoff relationship following prolonged drought. These catchments were mostly, high-relief, high-rainfall, well vegetated catchments, such as the one studied here and those typical of the east

coast of Australia. These results are reinforced here and shown through the propagation of drought through to vegetation and soil moisture and strong recovery following drought termination. This may be linked to droughts being less severe in the historical record compared to lower rainfall areas.

## 6 Conclusion

Global drought management strategies look to preserve and maintain resources under climate stress to promote quick

recovery once this stress is eased. Developing baseline data to understand different drought responses at the catchment scale is crucial for developing strategies that allow for this. Drought forecasting is generally poor across the globe, with many droughts only being identified after they have begun. This is further compounded by the potential for increased risk of drought under projected climate change. Due to this, a better characterization of how droughts occur, and end is required to ensure adequate preparation for long-term, severe drought events. The methods applied here identify drought and the wider

mechanisms associated with their cause and termination. This understanding is applicable in many environments due to the

recognized pattern of large-scale climate oscillations being associated with drought causation and smaller synoptic-scale events being associated with drought termination.

Various studies have identified drying trends in hydroclimate under climate change with an increased risk of protracted and more severe droughts occurring. Examples of the devastating impacts this may have are found in the
Millennium Drought, but also in the recent Texas and Californian droughts and ongoing crises in Africa. To mitigate these impacts, regions that experience high, reliable rainfall, as outlined here, could play a crucial role in ensuring water and food security in drought. While the scale of application is local within this study, the indices and remote sensed data can be applied almost anywhere in the world and at any scale. Importantly, assessing drought resilience at the local catchment scale may allow for better land use planning and resource management during drought both in the present and the future under climate
change. By quantifying response to drought in drought resilient areas, this study adds to the knowledge base required for this to occur.

**Author Contributions**

All coauthors conceptualized the paper and its scope. Authors DVK and AG designed the framework for analyses in this study, while data acquisition and processing was carried out by AG. Manuscript preparation was done by AG with
contribution from all coauthors.

**Code and Data Availability**

The SASMAS data is available on request from eng.newcastle.edu.au/sasmas/SASMAS/sasdata.html. Rainfall data used in this study is publicly available from the Australian Bureau of Meteorology from bom.gov.au, while NDVI data was obtained from reverb.echo.nasa.gov/reverb/

**Competing Interests**

The authors declare no conflict of interest.

**Acknowledgements**

The authors would like to thank the landholders of the Merriwa region for their hospitality and cooperation and acknowledge the work completed by those involved with the SASMAS project, especially the contribution of Dr. Tony Wells..
This work has been completed under an Australian Government Research Training Scholarship and CSIRO PhD top-up scholarship.



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



**Figure 1: Location of the study catchments with the Upper (red) and Lower (blue) Murray-Darling Basin catchments and NSW East Coast climate zone (green) also shown on inset which are referred to in the later analysis.**





**Figure 2: Diagram outlining the methods used to quantify drought from onset through to propagation and then termination.**




Figure 3: SPI6 and SWSI6 values for Merriwa (1908-2014) with periods of drought highlighted.





**Figure 4: Rates of drought onset (left) and termination (right) for the SPI6 (top) and SWSI6 (bottom).**

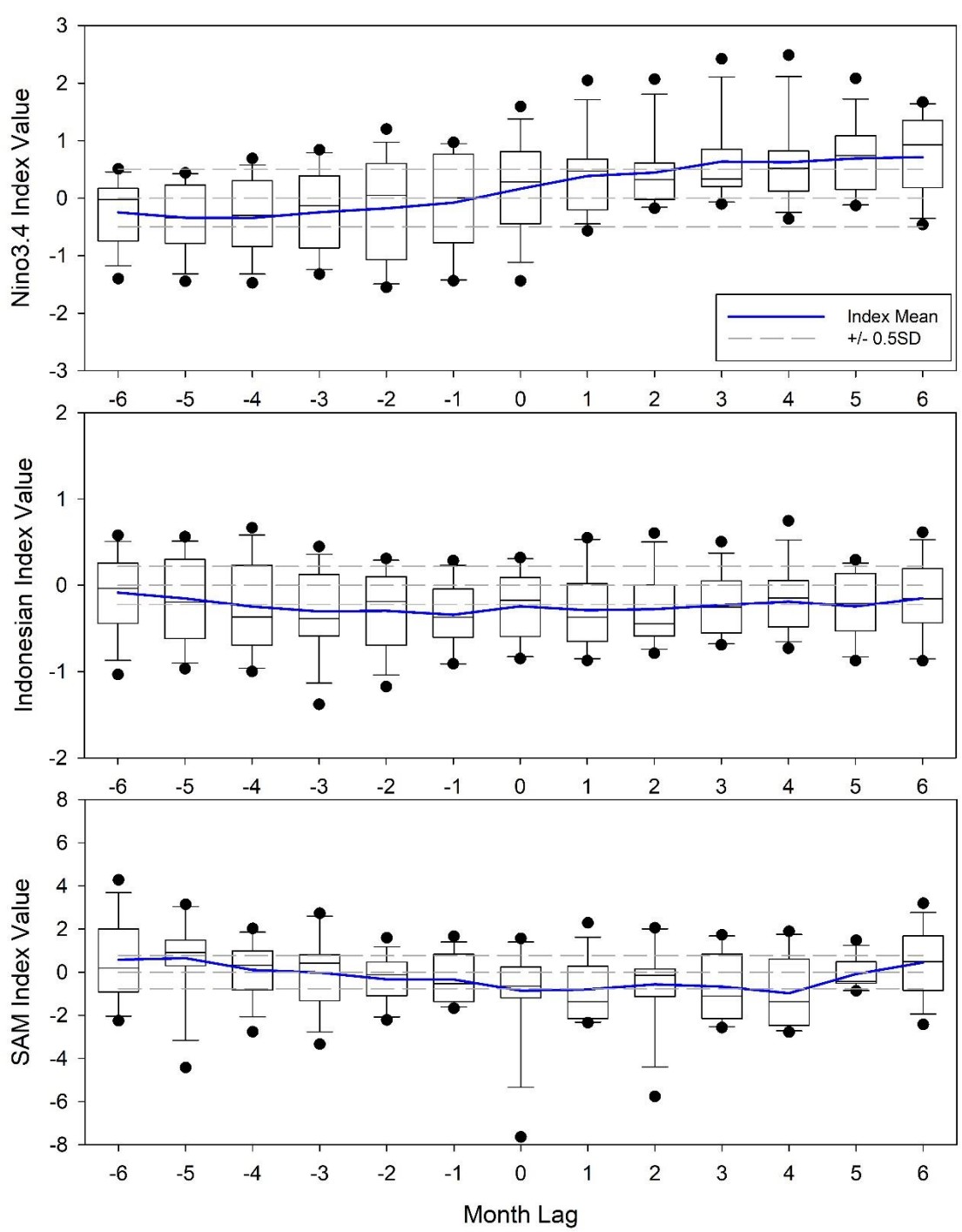

**Figure 5: Distribution of climate index values across the six months leading to and after drought initiation.**





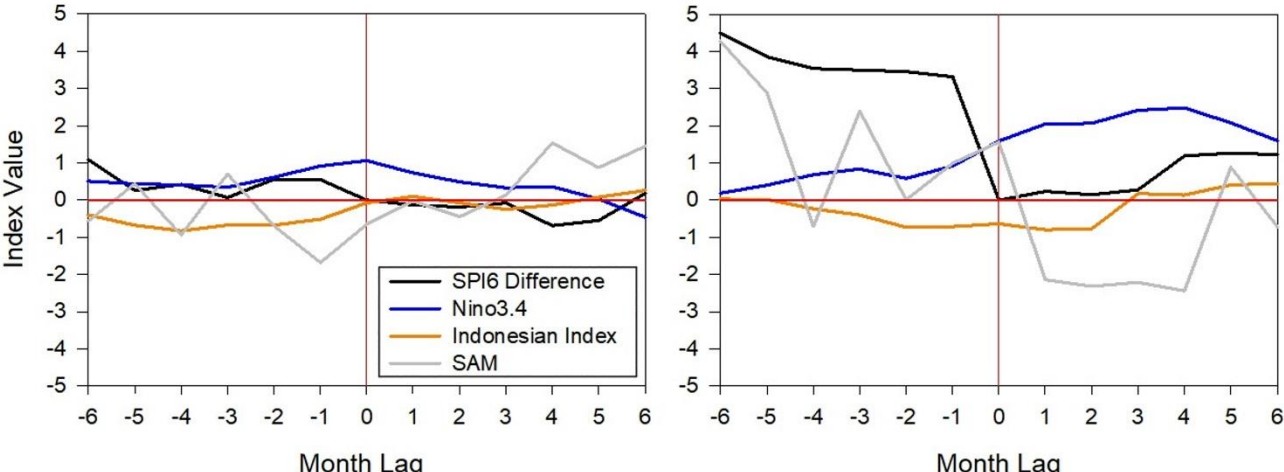

**Figure 6: Climate index values for Nino3.4, II and SAM and the SPI6 for the six-months prior to and post Drought onset for Droughts Four (left) and Nine (right).**





**Figure 7: Changes in SPI6 and monthly rainfall during drought termination for Droughts Four (top) and Nine (bottom).**


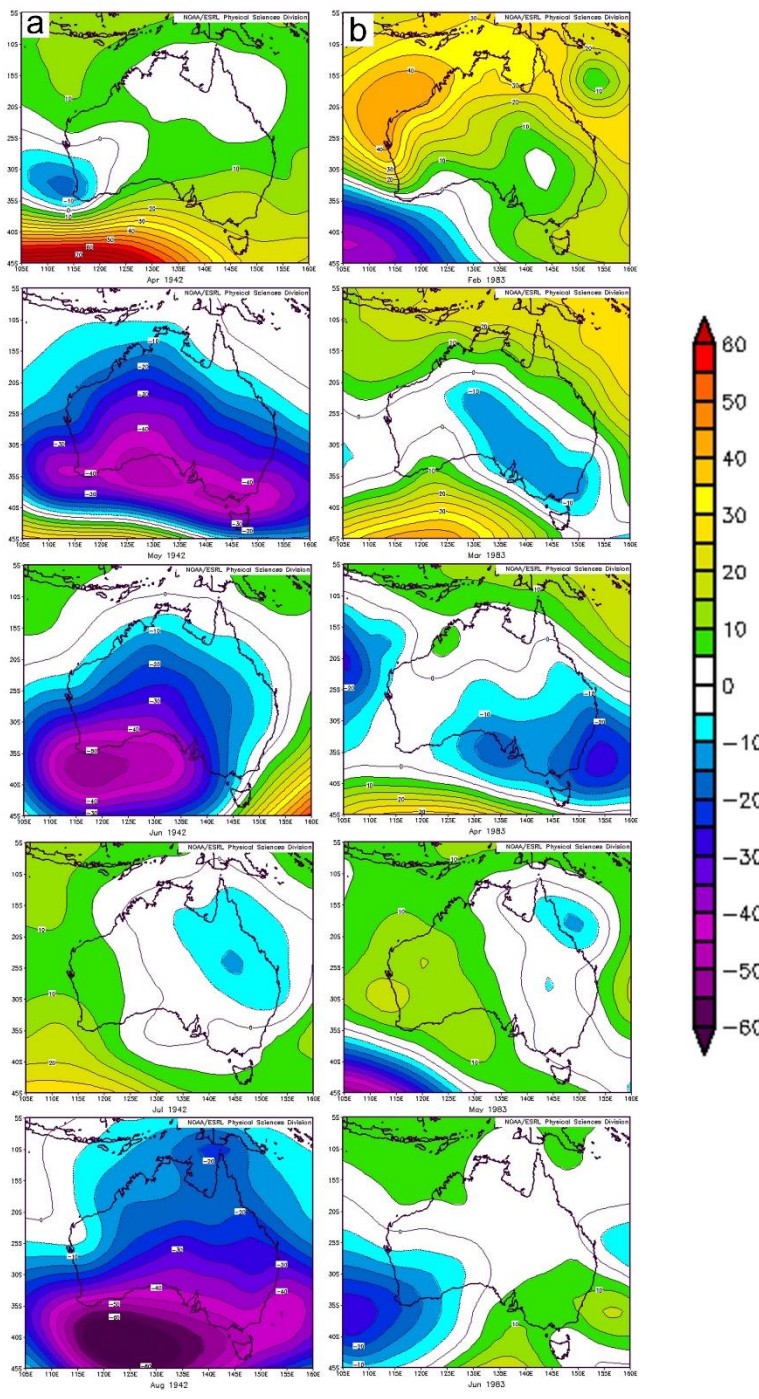

**Figure 8: Geopotential height (m) anomalies for the two months before and after drought onset for Droughts Four (a- April 1943 to August 1943 top to bottom) and Nine (b- February 1983 to June 1983 top to bottom).**



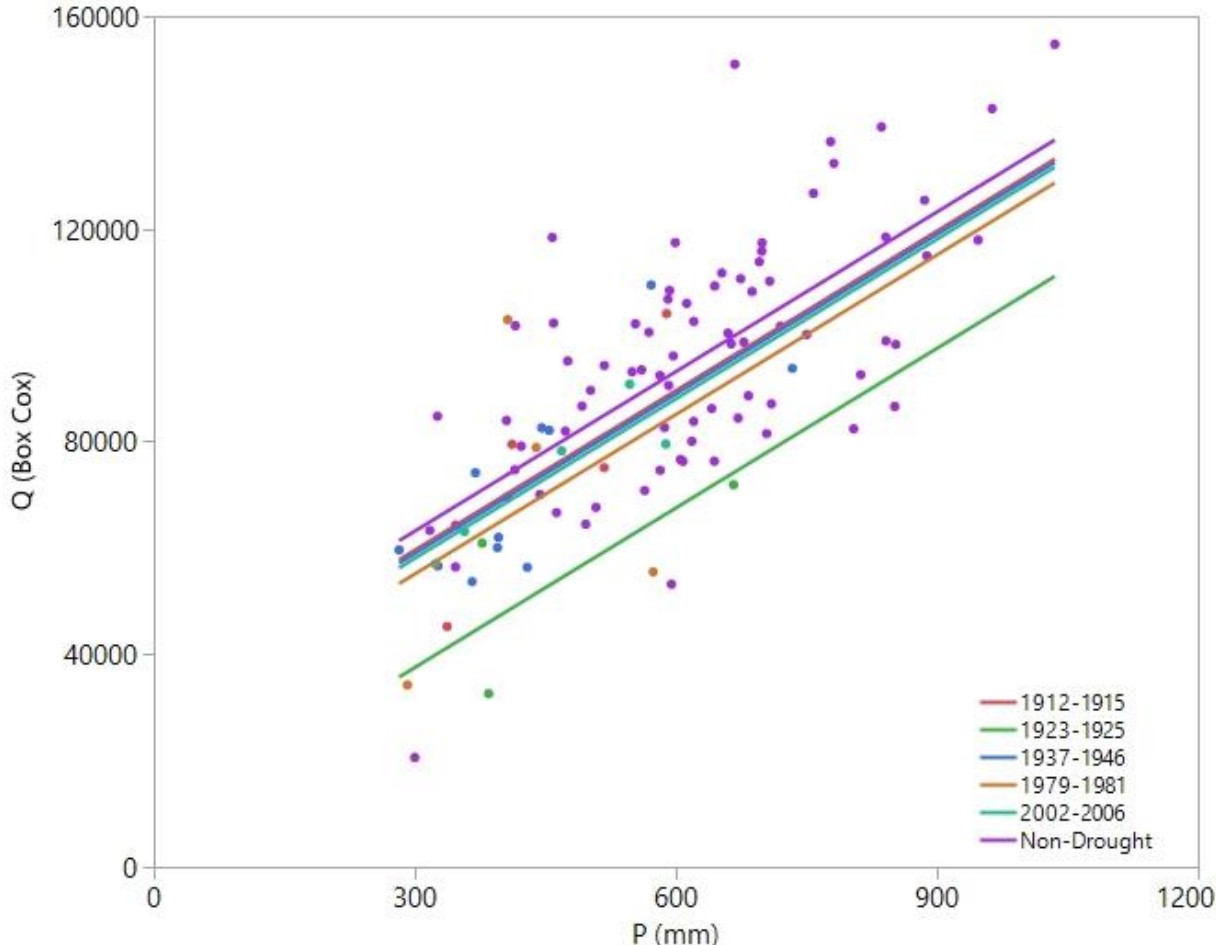

**Figure 9: Rainfall-runoff relationships for the drought periods and non-drought periods identified with the methods of Saft et al. (2015).**

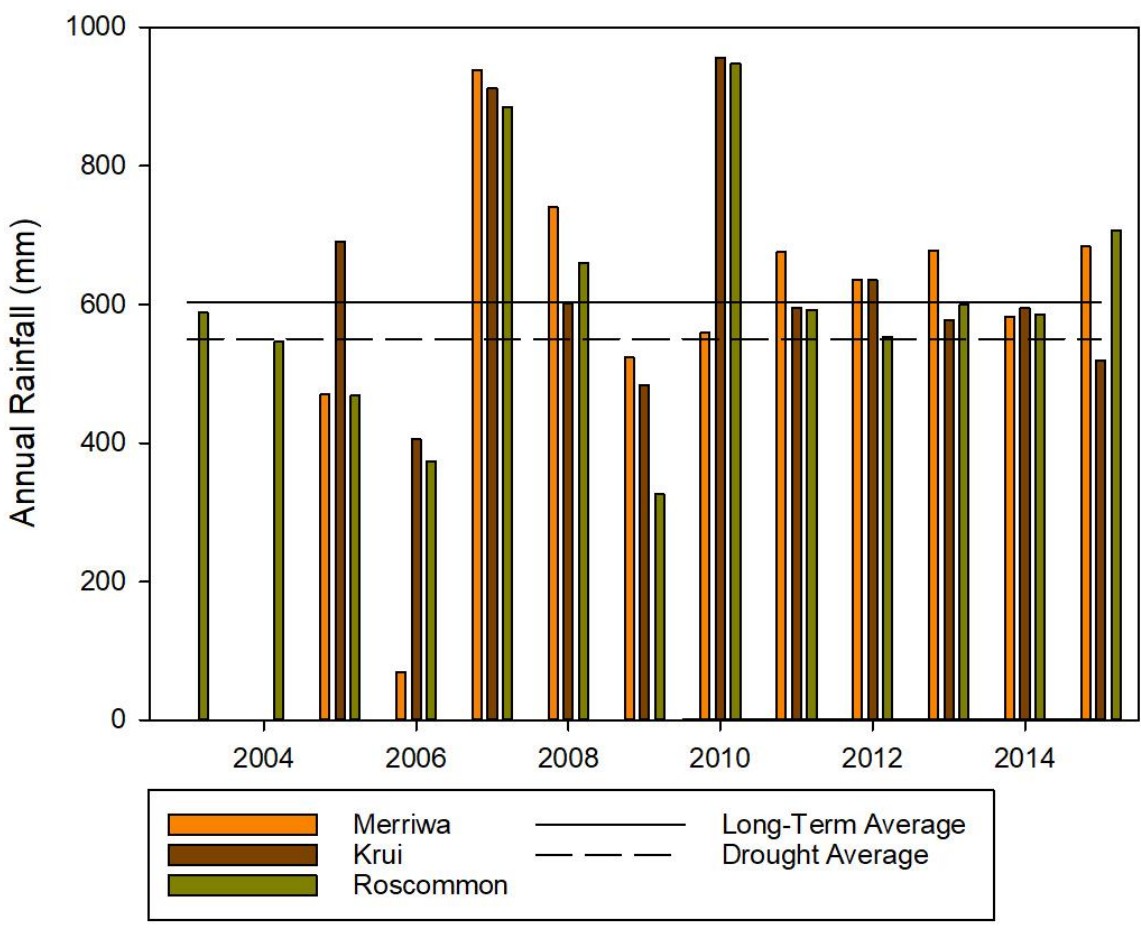

**Figure 10: Total annual rainfall for the Krui and Merriwa SASMAS stations (an average from all stations) and the BoM Roscommon station 2003-2015. N.B. Significant data missing from Merriwa rainfall totals in 2006 and 2010. Also presented are the long-term average for the Roscommon site and the drought period average across the three datasets.**



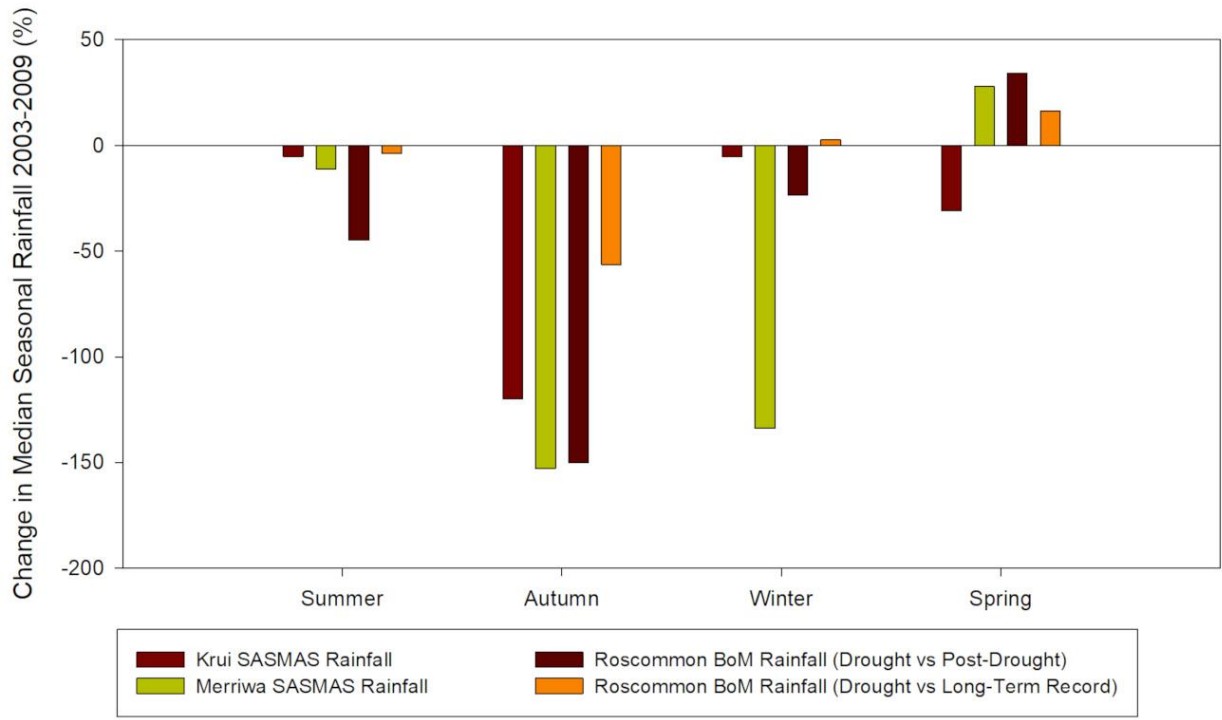

**Figure 11. Change in seasonal median rainfall (%) from the drought period (2003-2009) to the post drought period (2010-2015), and,**
**also between the drought period and the Roscommon long-term record (1970-2015).**





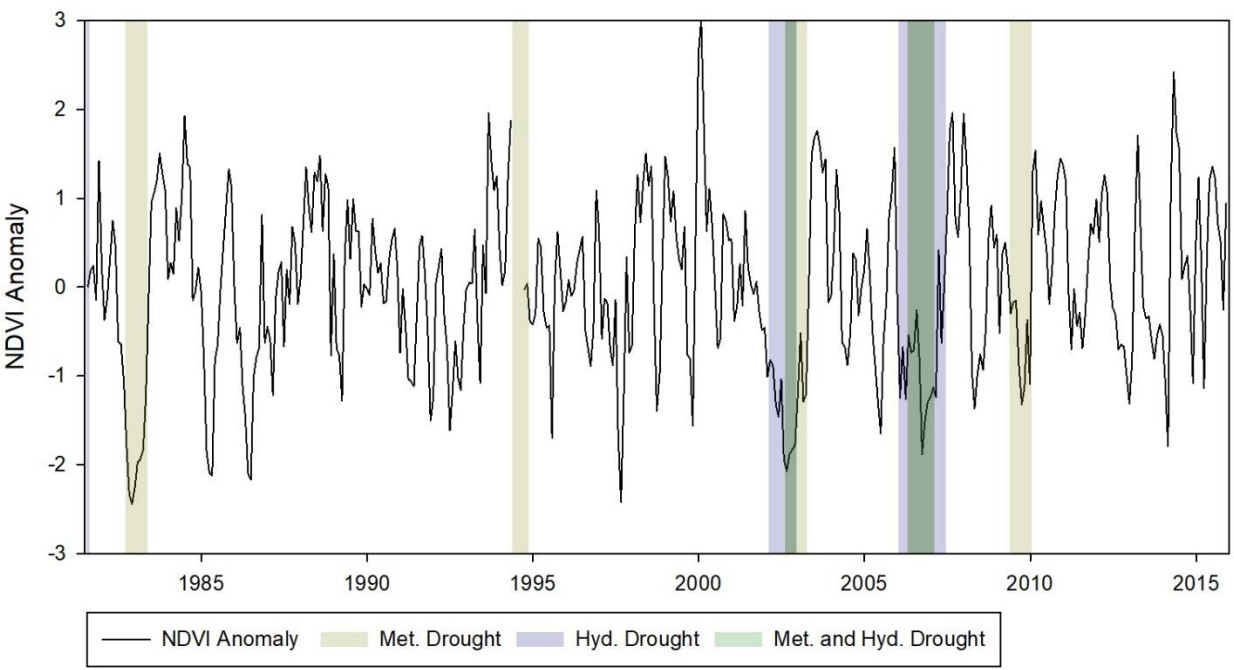

**Figure 12. Time-series AVHRR and MODIS NDVI anomaly for the Merriwa and Krui catchments.**






**Figure 13. SASMAS time-series soil moisture data, showing daily variation for the Krui (top) and Merriwa (bottom) Catchments at depths 0-300 mm, 300-600 mm, and 600-900 mm.**



**Table 1. Climate index data used in this study**

| Climate Mode | Index | Source | Description |
|---|---|---|---|
| ENSO | Nino3.4 | http://iridl.ldeo.columbia.edu/SOURCES/.Indices/.nino/.EXTENDED/ | Large scale ocean-atmosphere interactions within the tropical-subtropical Pacific Ocean that impact rainfall and temperatures globally. It has three phases; El Niño (indicated by values >0.5 cool SSTs in the east- reduced rainfall in eastern Aus.) neutral (values between 0.5 and -0.5) and La Niña (values < -0.5- warm SSTs in the east- increased rainfall in eastern Aus.) (Kaplan et al., 1998, Reynolds et al., 2002). |
| IOD | Indian Ocean SST Anomalies | http://www.jamstec.go.jp/frcgc/research/d1/iod/HTML/Dipole%20Mode%20Index.html | Variation in SST anomalies in the Indian Ocean is linked to variation in rainfall in countries local to the Indian Ocean, and, Europe and Nth and Sth America. Warm SST anomalies, represented by the Indonesian Index (II), to the NW of Aus. are linked with high rainfall across the SE of the continent (Verdon and Franks 2005). |
| SAM | Southern Annular Mode | http://www.cpc.ncep.noaa.gov/products/precip/CWlink/daily_ao_index/aao/aao.shtml  http://research.jisao.washington.edu/data_sets/aao/slp/ | SAM, or the Antarctic Oscillation (AAO) is characterised by temperature differences between the tropics and southern polar regions which leads to changes in westerly winds that either constrain polar vortexes or allow them to pass over southern Australia. In its positive phase, these vortexes are constrained to south, resulting in dry conditions during autumn-winter and rainfall in spring-summer in SE Aus. During the negative phase, storm fronts bring autumn-winter rainfall but dry conditions during spring-summer (Verdon-Kidd et al., 2017). |






**Table 2. Summary of meteorological drought events as identified using the SPI6.**

| Drought No. | Duration | Length | Average SPI6 | SPI6 Break |
|---|---|---|---|---|
| 1 | Jan 19- May 20 | 17 | -1.37 | 2.04 |
| 2 | Feb 23- July 23 | 6 | -1.4 | 0.4 |
| 3 | Oct 39- Nov 40 | 14 | -1.56 | 0.99 |
| 4 | Dec 41- May 42 | 6 | -1.42 | 0.73 |
| 5 | Oct 46- Jun 47 | 9 | -1.78 | 0.51 |
| 6 | Jan 57- Dec 57 | 12 | -1.28 | 0.81 |
| 7 | Feb 65- Dec 65 | 8 | -2.11 | 0.91 |
| 8 | Dec 79- April 81 | 17 | -1.43 | 0.71 |
| 9 | Sept 82-April 83 | 8 | -2.32 | 1.29 |
| 10 | May 94- October 94 | 6 | -1.8 | 1.72 |
| 11 | Aug 02- March 03 | 8 | -1.41 | 0.84 |
| 12 | April 06- Jan 07 | 10 | -1.35 | 0.9 |
| 13 | May 09- Dec 09 | 8 | -1.27 | 0.78 |
| Average | | 10 | -1.36 | 0.97 |





**Table 3. Summary of hydrological drought events as identified using the SWSI6.**

| Drought No. | Duration | Length | Average SWSI6 | SWSI6 Break |
|:---:|:---:|:---:|:---:|:---:|
| 1 | Feb 19- Dec 19 | 11 | -2.01 | 1.64 |
| 2 | Jan 23- Aug 23 | 8 | -1.19 | 0.9 |
| 3 | Sept 39- Mar 40 | 7 | -1.45 | 0.9 |
| 4 | Jul 41- Jun 42 | 12 | -1.75 | 2.53 |
| 5 | March 65- Sept 65 | 7 | -1.45 | 0.81 |
| 6 | Feb 80- Jul 81 | 17 | -1.65 | 0.14 |
| 7 | Feb 02- Nov 02 | 10 | -1.43 | 1.47 |
| 8 | Jan 06- May 07 | 17 | -1.06 | 3.29 |
| Average | | 12 | -1.50 | 1.46 |