# Peer review of "Catchment-scale drought: capturing the whole drought cycle using multiple indicators"

_Hydrology and Earth System Sciences, 2019_

## Referee Comment (RC1) · Anonymous Referee #1 · 23 Aug 2019

In this study, the authors quantify drought within a single catchment in eastern Australia using a variety of indicators to look at how drought is expressed across the landscape. They also analyze the dominant causes of both drought onset (attributable primarily to various important climate modes) and termination (via low pressure troughs). They further note that drought recovery is rapid, with little evidence for prolonged effects (e.g., on vegetation or runoff) during the post drought periods. This is a detailed and interesting study of drought within a highly localized area. I am happy to recommend acceptance, assuming the authors can address my relatively minor criticisms centered around some of the analysis choices.

(*) The description and application of the NDVI analysis is really quite thin. What is the dominant land cover type in this catchment (forest, shrub, agriculture, etc)? How

extensive is human land use (agriculture or pasture)? What is the seasonality of the vegetation, and when in the year is peak productivity and vegetation growth? This is all important information that is required to contextual how drought, and other stress factors, affect vegetation in this catchment. Additionally, the reverb link on line 155 does not work. As I also mention later, I think the NDVI analysis would further benefit from explicit separation of the warm (growing?) vs cold (dormant?) season droughts.

(*) Line 193: what are the units on these drought onset/termination rates? Are these changes in the index value (e.g., SPI6) per month?

(*) I found the climate index discussion to be a bit confusing, primarily because the analysis of drought conflates events during both the warm and cold season. The problem, as the authors even admit, is that the influence of climate modes in this region is highly seasonal. Seasonality is mentioned, but it is hand waved away (line 215) for a simplistic statement that Indian ocean and SAM cause drought and El Nino sustains it. Further, the lack of a strong pre-drought signal in indices (especially ENSO) could just be a function of the long-term drought index being used (integrating over 6 months) or the fact that this catchment responds quickly to these modes with little time lag. It would be useful, I think, for the authors to a priori separate cold and warm season drought events and THEN conduct the comparisons with the climate modes. As currently written, I just find this section to be a bit disorganized.

(*) For readers who are not familiar, it would be good for the authors to explicitly define "rainfall-runoff relationship" and why changes in it might matter (or what they would mean).

---

## Referee Comment (RC2) · Anonymous Referee #2 · 6 Jan 2020

This study provides analysis on droughts propagation from atmosphere to different terrestrial compartments in two (east) Australian catchments since the beginning of the 20th Century. The analysis performed is quite comprehensive and detailed for the recent (Millenium) drought event - touching the different aspects of the droughts including the atmospheric drivers. This is a very valuable contribution and I would recommend for its publication in HESS. I have some minor comments/clarifications, which I assume the authors would easily handle.

1. Somehow I missed the information on how the authors have objectively defined the criteria for the drought onset and termination.

2. While there appears to be two study catchments analyzed in this study – but the hydrological droughts (SWSI6) and NDVI anomaly (in Figures 3, 4, 9 and 12) is just a

single plot. For which catchments these data refer to? Or these plots use data combined for both basins – in this case how the underlying drought indices were aggregated into single values?

3. I understand that the authors used the AWAP simulated streamflows in their analyses. Since the following hydrological analysis is based on this modeled dataset, I would recommend the authors to make a quality check (skill assessment) against the available observed streamflow – though it might be the short time series – in my opinion the analysis will provide good foundation.

Related to the above – I would also recommend the authors to check the differences between the precipitation datasets (and the resulting meteorological drought index) i.e., one use as forcing in the AWAP product and the one the authors used in their analysis (i.e., BoM-Roscommon). This is really important to check in light of the author's discussion/conclusion on Page 10: "Not all meteorological droughts were found to progress to hydrological drought for our study catchment. The best example of this is the absence of a hydrological drought with the 1982– 83 meteorological event. This was the most severe meteorological drought, with an average SPI6 of -2.32, and had a rapid rate of onset (outlier in Fig. 3), however, there was no associated hydrological drought."

4. Could the authors explicitly specify the motivation as well the settings (parameters) they used in the Box-Cox transformation of discharge. What is the unit of Q (Y-axis) in Figure 9.

5. It is not clear which line on the Figure 4 corresponds to drought #4 or #9 (as mentioned many times in the manuscript). I can only guess.

6. Line 224: Please explain how did you identify the specific season from information provided in Figure 5.

7. I do not concur with the author's interpretation (Lines 270, 280, and 347) "the propagation from meteorological drought to agricultural drought is rapid. . . " Just because

the lag is zero it does not mean a rapid propagation. Note that you have taken SPI6, which accounts for the past 6 months of (accumulated) precipitation anomaly – which inherently account for the antecedent conditions (and create a memory effect). I would like to hear the authors opinion on this issue.

8. Since soil moisture also exhibits strong seasonality, I would have expected that authors to remove those seasonal effect (as they consider in case of NDVI) and consider the anomaly term in their analysis. Please comment on this.

---

## Author Response (AR2)

Earth Science Building
School of Environmental and Life Sciences
University of Newcastle, NSW, 2308, Australia
Email: abraham.gibson@uon.edu.au

Dear Prof. Romano,

The paper "*Catchment-scale drought: capturing the whole drought cycle using multiple indicators*" (Manuscript ID: hess-2019-311) authored by Abraham Gibson, Danielle Verdon-Kidd, Greg Hancock and Gary Willgoose has been revised as per the reviewers' comments.

We sincerely thank you as the editor, and the two reviewers for providing detailed, constructive, and extremely helpful comments. We have carefully reviewed and responded to each comment. Please find these responses enclosed along with a list of amendments and a revised version of the manuscript with changes marked in red.

We look forward to the outcome of your assessment in due course.

Yours sincerely,

[Figure]

**Abraham Gibson**
PhD Candidate
School of Environmental and Life Sciences
The University of Newcastle

THE UNIVERSITY OF
NEWCASTLE
AUSTRALIA

**Response to Anonymous Referee #1 on "Catchment-scale drought: capturing the whole drought cycle using multiple indicator" by A.J. Gibson et al.**

**Review Comment 1:** *The description and application of the NDVI analysis is really quite thin. What is the dominant land cover type in this catchment (forest, shrub, agriculture, etc)? How extensive is human land use (agriculture or pasture)? What is the seasonality of the vegetation, and when in the year is peak productivity and vegetation growth? This is all important information that is required to contextual how drought, and other stress factors, affect vegetation in this catchment. Additionally, the reverb link on line 155 does not work. As I also mention later, I think the NDVI analysis would further benefit from explicit separation of the warm (growing?) vs cold (dormant?) season droughts.*

**Authors' Response:** The dominant land use and cover types are outlined in lines 84–87. The seasonality of vegetation growth is an important aspect of agricultural drought that was overlooked in the site description by the authors. This has now been included in the revised manuscript; lines 84–87:

"the area is of agricultural significance and sustains grazing and cropping, making up around 75% of each catchment. (Hancock et al., 2015; Rudiger et al., 2007). Vegetation is described as cover grasses with sparse eucalypt forest in the lower and central regions of the catchments, while dense, wet sclerophyll forest dominates the northern slopes (Kunkel et al., 2019). Vegetation cover is generally consistent throughout the year due to an even distribution in rainfall."

The separation of warm and cool season droughts is addressed in Comment Three.

**Review Comment 2:** *Line 193: what are the units on these drought onset/termination rates? Are these changes in the index value (e.g., SPI6) per month?*

**Authors' Response:** These units are index value per month, these have been added.

**Review Comment 3:** *I found the climate index discussion to be a bit confusing, primarily because the analysis of drought conflates events during both the warm and cold season. The problem, as the authors even admit, is that the influence of climate modes in this region is highly seasonal. Seasonality is mentioned, but it is hand waved away (line 215) for a simplistic statement that Indian ocean and SAM cause drought and El Nino sustains it. Further, the lack*

*of a strong pre-drought signal in indices (especially ENSO) could just be a function of the long-term drought index being used (integrating over 6 months) or the fact that this catchment responds quickly to these modes with little time lag. It would be useful, I think, for the authors to a priori separate cold and warm season drought events and THEN conduct the comparisons with the climate modes. As currently written, I just find this section to be a bit disorganized.*

**Authors' Response:** We acknowledge that some studies examine drought (especially flash drought) as cold or warm season droughts, however this study aimed to look at how meteorological droughts are sustained over multiple seasons and then propagate through to hydrological and agricultural droughts. That is, the drought events studied in this paper tend to span both cool and warms seasons and are hence some of the most impactful. In Figure 5, we identified that across all droughts in the approx. 100-year record, there is a trend in cool Indian Ocean sea surface temperatures (SSTs) combined with an increasing El Niño Southern Oscillation (ENSO) index leading to drought initiation. Following this, El Niño conditions during the months after drought initiation continue to suppress rainfall, leading to sustained drought. This is indicated by most droughts beginning in summer with the Indian Ocean modes supressing winter and spring rainfall and ENSO further suppressing rainfall (see figure below). This combines effect leads. The role of the Indian Ocean modes and ENSO in causing drought in southeast Australia (especially on the east coast) is much debated, and a key finding here is that long-term droughts are caused by the interactions of both, along with the Southern Annular Mode (Lines 199–214). We also acknowledge that the seasonality of the climate modes is a key mechanism in causing long-term drought. When the climate modes are in their "dry-phase" the sequential nature of their suppression of rainfall causes sustained drought (Lines 210–215).

[Figure]

Figure 1: Onset months of droughts from the SPI6 record grouped by season (data from Table 1 in manuscript).

**Review Comment 4:** *For readers who are not familiar, it would be good for the authors to explicitly define "rainfall-runoff relationship" and why changes in it might matter (or what they would mean).*

**Authors' Response:** The authors thank the reviewer for this suggestion and have clarified this in Sect. 3.4 (Lines 137–140):

"Persistent changes in the rainfall-runoff relationship have been observed after the Millennium Drought by Saft et al. (2015). This relationship is the amount of runoff generated by a given amount of rainfall. Persistent changes in this indicate a change in hydrology in a catchment; indicating that the catchment has undergone a sustained change or has been unable to recover from drought (Saft et al., 2015)."

**Response to Anonymous Referee #2 on "Catchment-scale drought: capturing the whole drought cycle using multiple indicator" by A.J. Gibson et al.**

**Review Comment 1:** *Somehow, I missed the information on how the authors have objectively defined the criteria for the drought onset and termination.*

**Authors' Response:** The authors refer the reviewer to the definitions of drought onset and termination in line 121. In particular:

"For this study, an SPI period of 6 months (SPI6) was chosen to capture persistent droughts, with meteorological drought onset being defined as a period of six successive months of SPI6 below -1 (Dettinger 2013; Verdon-Kidd et al., 2017). Drought severity was then categorized, based on SPI values, as; mild (-1 < SPI < -0.5), moderate (-1.5 < SPI < -1), severe (-2 < SPI < -1.5) and extreme (SPI < -2), as recommended by McKee et al., (1993, 1995).  Following this, termination of a drought event was then defined as a period of six, successive months where SPI6 was above -1 (Dettinger, 2013; Verdon-Kidd et al., 2017)."

**Review Comment 2:** *While there appears to be two study catchments analyzed in this study – but the hydrological droughts (SWSI6) and NDVI anomaly (in Figures 3, 4, 9 and 12) is just a single plot. For which catchments these data refer to? Or these plots use data combined for both basins – in this case how the underlying drought indices were aggregated into single values?*

**Authors' Response:** The data refers to a single record created using average values across both catchments. This was done to match the single rainfall record developed from the two BoM stations (Line 112). A single record of streamflow and NDVI were developed first by taking the average across both catchments, with drought indices calculated from this. Line 156 has been modified to clarify this. The data then diverges to both catchments for the Millennium Drought study due to an increase in data availability.

**Comment 3:** *I understand that the authors used the AWAP simulated streamflows in their analyses. Since the following hydrological analysis is based on this modeled dataset, I would recommend the authors to make a quality check (skill assessment) against the available observed streamflow – though it might be the short time series – in my opinion the analysis will provide good foundation. Related to the above – I would also recommend the authors to check the differences between the precipitation datasets (and the resulting meteorological drought index) i.e., one use as forcing in the AWAP product and the one the authors used in their analysis (i.e., BoM-Roscommon). This is really important to check in light of the author's discussion/conclusion on Page 10: "Not all meteorological droughts were found to progress to hydrological drought for our study catchment. The best example of this is the absence of a hydrological drought with the 1982–83 meteorological event. This was the most severe meteorological drought, with an average SPI6 of -2.32, and had a rapid rate of onset (outlier in Fig. 3), however, there was no associated hydrological drought."*

**Authors' Response:** The authors thank the reviewer for their comment and have added a reference to Gibson (2016) to line 112 of the revised paper which details the instrumental validation for this catchment suggested by the reviewer. Demonstrated in this reference is a local validation of the AWAP streamflow against the limited available observed data. The authors have compared the drought indices derived from the AWAP rainfall product. The two SPI records were found to be near identical, with the AWAP SPI = 0.94 x Observed SPI ($R^2$ = 0.91, p < 0.01). Between the two records there is only a slight difference in the timing of droughts. As the paper is already lengthy, this analysis was not explicitly included, however we agree this should be cited.

**Review Comment 4:** *Could the authors explicitly specify the motivation as well the settings (parameters) they used in the Box-Cox transformation of discharge. What is the unit of Q (Y-axis) in Figure 9?*

**Authors' Response:** This transformation results in the heavily skewed runoff data to approximate a normal distribution, with the relationship with rainfall then becoming linear and more widely applicable as outlined in Saft et al. (2015). This has been added to line 141 of the revised manuscript to clarify. The parameters used have been explained further in the methods section. As this is a transformation of values, Q becomes unitless.

"This transformation results in the heavily skewed runoff data to approximate a normal distribution, with the relationship with rainfall then becoming linear and more widely applicable (Saft et al., 2015). and regressed against annual rainfall. This was carried out for each "drought period" and non-drought years using the best transformation selection method, with $\lambda$ = 0.264. A t-test was used to determine if a significant change within this relationship had occurred within each drought period (Saft et al., 2015)."

**Review Comment 5:** *It is not clear which line on the Figure 4 corresponds to drought #4 or #9 (as mentioned many times in the manuscript). I can only guess.*

**Authors' Response:** We thank the reviewer for pointing out this oversight and in response have adjusted lines 217-223 accordingly. Rather than focusing on Figure 4, we have reworded the text to highlight that Drought 4 is a typical onset, while Drought 9 is more rapid and an outlier in the drought record.

**Review Comment 6:** *Line 224: Please explain how did you identify the specific season from information provided in Figure 5.*

**Authors' Response:** The seasonality of the droughts is clarified by adjusting line 217 and the caption to Figure 6. The reference to Figure 5 in line 224 has also been changed to Figure 6.

**Review Comment 7:** *I do not concur with the author's interpretation (Lines 270, 280, and 347) "the propagation from meteorological drought to agricultural drought is rapid. . . " Just because the lag is zero it does not mean a rapid propagation. Note that you have taken SPI6, which accounts for the past 6 months of (accumulated) precipitation anomaly – which inherently account for the antecedent conditions (and create a memory effect). I would like to hear the authors opinion on this issue.*

**Authors' Response:** This is an interesting point and understanding catchment memory's role in drought is a topic widely discussed in the literature. While memory would have a role in drought propagation here, the lack of a lag in correlation between NDVI and SPI is unique in this study when compared to others that show lags of up to 5 months (using SPI6) improving correlations (e.g. Verdon-Kidd et al.,. 2017). As a result of this, we have come to the conclusion that the transition from meteorological to agricultural drought is relatively quicker in this catchment compared to other areas. References and edits to line 284 have been made to clarify;

"This shows the propagation from meteorological drought to agricultural drought is rapid compared to other studies (e.g. Verdon-Kidd et al., 2017). This is an important aspect of using the SPI6 to quantify drought here. A response at short timescales between rainfall and vegetation may not be seen, however using the longer-term SPI highlights the response of vegetation after sustained rainfall deficits (Dettinger et al., 2013, Verdon-Kidd et al., 2017)."

The role of catchment memory, size, vegetation and rainfall patterns does warrant further investigations, and we hope to in the future, however this is just not possible within the current scope of this paper.

**Review Comment 8:** *Since soil moisture also exhibits strong seasonality, I would have expected that authors to remove those seasonal effect (as they consider in case of NDVI) and consider the anomaly term in their analysis. Please comment on this.*

**Authors' Response:** The authors recognise that there can be strong seasonal variations in soil moisture, however, at our particular study site, rainfall is quite uniform throughout the year (compared to other areas of Australia with a distinct seasonality in rainfall). Additionally, volumetric soil moisture data is important for understanding the water available to plants during drought and standardising the data would remove this aspect. Standardising the three depths of soil moisture also removes the differences in soil moisture between them. The fact there is small variation in deep soil moisture over the time series is believed to be important for catchment recovery to drought. While both the raw timeseries and anomalies could be explored, this would extend the paper with adding little to the discussion that is not already captured in the raw time series (as show in Fig 2).

[Figure]

Figure 2: Comparison of anomaly and raw soil moisture data for the Krui catchment. The Merriwa was not included due to the similarities between the datasets.

**List of Changes**

1. Line 84-87: added; "Due to the deep and fertile soil profiles, the area is of agricultural significance and sustains grazing and cropping; making up around 75% of each catchment. (Hancock et al., 2015; Rudiger et al., 2007). Vegetation is described as cover grasses with sparse eucalypt forest in the lower and central regions of the catchments, while dense, wet sclerophyll forest dominates the northern slopes (Kunkel et al., 2019). Vegetation cover is generally consistent throughout the year due to an even distribution in rainfall."

2. Line 114: added; "and a local validation in Gibson (2016)."

3. Line 123: added; "(Dettinger 2013; Verdon-Kidd et al., 2017)."

4. Line 125-127: changed "Following Verdon-Kidd et al (2017), termination of a drought event was then defined as a period of six, successive months where SPI6 was above -1 (McKee et al., 1993; McKee et al., 1995). Rainfall data from the composite record of the Roscommon and Terragong weather stations was used for calculation of the SPI." to "Following this, termination of a drought event was then defined as a period of six, successive months where SPI6 was above -1 (Dettinger, 2013; Verdon-Kidd et al., 2017). Rainfall data from the composite record of the Roscommon and Terragong weather stations was used for calculation of the SPI.

5. Lines 138-141: added; "Persistent changes in the rainfall-runoff relationship have been observed after the Millennium Drought by Saft et al. (2015). This relationship is the amount of runoff generated by a given amount of rainfall. Persistent changes in this indicate a change in hydrology in a catchment; indicating that the catchment has undergone a sustained change or has been unable to recover from drought (Saft et al., 2015)." and removed "Changes in the rainfall-runoff relationship during drought were examined by methods similar to those presented in Saft et al. (2015)."

6. Lines 147-151: added; "This transformation results in the heavily skewed runoff data to approximate a normal distribution, with the relationship with rainfall then becoming linear and more widely applicable (Saft et al., 2015). This was carried out for each "drought period" and non-drought years using the best transformation selection method, with $\lambda = 0.264$." Removed "and regressed against annual rainfall" from Line 149.

7. Lines 163-164: added; " To match the single rainfall and streamflow records, an average was calculated across both catchments."

8. Lines 203-204: added; "SPI units.month$^{-1}$" in four locations in line with review comment.

9. Lines 226-227: added dates; "Four (Dec 1941–May 1942) and Nine (Sept 1982–Apr 1983)" after droughts to show when they occurred. Changed; "Drought Four represents a typical drought onset, as depicted in Fig. 4." to Drought Four represents a drought onset similar to the majority presented in Fig. 4"

10. Lines 233-235: changed; "…in terms of onset and can be easily seen as an outlier in the SPI6 drought onset rates in Fig. 4." to "…in terms of onset with a more rapid rate of onset compared to the mean rate in rates in Fig. 4."

11. Line 234: changed "Fig 5" to "Fig 6b".

12. Lines 284-287: added; "compared to other studies (e.g. Verdon-Kidd et al., 2017). This is an important aspect of using the SPI6 to quantify drought here. A response at short timescales between rainfall and vegetation may not be seen, however using the longer-term SPI highlights the response of vegetation after sustained rainfall deficits (Dettinger et al., 2013, Verdon-Kidd et al., 2017)."

13. Lines 442-443: added reference; "Gibson, A.J. (2016). Estimating Sediment Yield From a Large, Unguaged South-East Australian Catchment. Hons Thesis. The University of Newcastle, Australia."

14. Line 541: Figure 6 caption changed from; "Figure 6: Climate index values for Nino3.4, II and SAM and the SPI6 for the six-months prior to and post Drought onset for Droughts Four and Nine." to "Figure 6: Climate index values for Nino3.4, II and SAM and the SPI6 for the six-months prior to and post Drought onset for Droughts Four (onset at Dec 1941; left) and Nine (onset at Sept 1982; right)."